# Citric Acid Enhances Plant Growth, Photosynthesis, and Phytoextraction of Lead by Alleviating the Oxidative Stress in Castor Beans

**DOI:** 10.3390/plants8110525

**Published:** 2019-11-19

**Authors:** Zahid Imran Mallhi, Muhammad Rizwan, Asim Mansha, Qasim Ali, Sadia Asim, Shafaqat Ali, Afzal Hussain, Salman H. Alrokayan, Haseeb A. Khan, Pravej Alam, Parvaiz Ahmad

**Affiliations:** 1Department of Environmental Science and Engineering, Government College University, Faisalabad 38000, Pakistan; zahid.mallhi@yahoo.com (Z.I.M.); mrazi1532@yahoo.com (M.R.); afzaalh345@gmail.com (A.H.); 2Department of Chemistry, Government College University, Faisalabad 38000, Pakistan; 3Department of Botany, Government College University, Faisalabad 38000, Pakistan; qasimbot_uaf@yahoo.com; 4Department of Chemistry, Government College Women University, Faisalabad 38000, Pakistan; sadia1asim@gmail.com; 5Department of Biological Sciences and Technology, China Medical University (CMU), Taichung City 40402, Taiwan; 6Department of Biochemistry, College of Science, King Saud University, Riyadh 11451, Saudi Arabia; salrokayan@ksu.edu.sa (S.H.A.); Haseeb_khan@yahoo.com (H.A.K.); 7Biology Department, College of Science and Humanities, Prince Sattam bin Abdulaziz University (PSAU), Alkharj 11942, Saudi Arabia; ps.alam@psau.edu.sa; 8Botany and Microbiology Department, College of Science, King Saud University, Riyadh 11451, Saudi Arabia; parvaizbot@yahoo.com

**Keywords:** lead, castor beans, citric acid, phytoextraction, antioxidant enzyme

## Abstract

Lead (Pb) toxicity has a great impact in terms of toxicity towards living organisms as it severely affects crop growth, yield, and food security; thus, warranting appropriate measures for the remediation of Pb polluted soils. Phytoextraction of heavy metals (HMs) using tolerant plants along with organic chelators has gained global attention. Thus, this study examines the possible influence of citric acid (CA) on unveiling the potential phytoextraction of Pb by using castor beans. For this purpose, different levels of Pb (0, 300, 600 mg kg^−1^ of soil) and CA (0, 2.5, and 5 mM) were supplied alone and in all possible combinations. The results indicate that elevated levels of Pb (especially 600 mg kg^−1^ soil) induce oxidative stress, including hydrogen peroxide (H_2_O_2_) and malanodialdehyde (MDA) production in plants. The Pb stress reduces the photosynthetic traits (chlorophyll and gas exchange parameters) in the tissues of plants (leaves and roots), which ultimately lead to a reduction in growth as well as biomass. Enzyme activities such as guaiacol peroxidase, superoxide dismutase, ascorbate peroxidase, and catalase are also linearly increased in a dose-dependent manner under Pb stress. The exogenous application of CA reduced the Pb toxicity in plants by improving photosynthesis and, ultimately, plant growth. The upsurge in antioxidants against oxidative stress shows the potential of CA-treated castor beans plants to counteract stress injuries by lowering H_2_O_2_ and MDA levels. From the results of this study, it can be concluded that CA treatments play a promising role in increasing the uptake of Pb and reducing its phytotoxicity. These outcomes recommend that CA application could be an effective approach for the phytoextraction of Pb from polluted soils by growing castor beans.

## 1. Introduction

Heavy metals have been of much interest to researchers and scientists regarding environmental safety, and among this, lead (Pb) has gained substantial consideration as a persuasive environmental hazard. Both natural and anthropogenic activities such as the disposal of municipal sewage sludge, fertilizer application, practices of mines, forest fires, industrial fumes, storage batteries such as lithium ion batteries, volcanic eruptions, igneous rocks, ores smelting, paints, gasoline, and explosives are the major contributors to the release of Pb in the external environment [1]. Lead has been known as an inorganic toxin (cannot be degraded) even at lower concentrations and is readily absorbed in cultivated soils, which is easily taken up by various organs of plants. The major levels of Pb exhibit in the food chain through accumulation/absorption impose severe threat towards human health [2]. In the plant environment, Pb phytotoxicity results in a stop to plant growth, depression of seed germination, disruption of cellular structures, impairment of photosynthesis, imbalance of hormones, ion homeostasis, and the over-generation of reactive oxygen species (ROS) as well as inhibition of enzyme activities [3]. As a result, crops cultivated in these Pb-polluted soils directly affect agricultural production. Therefore, it is essential to remediate Pb contamination in the soil–plant environment. The use of green-plants to remediate contaminated soils is a long term, cost-effective, and eco-friendly tool [4,5]. However, the plants’ ability to uptake and translocate HMs into upper harvestable parts are dependent on soil type, plant species, and environmental conditions [6,7,8]. In recent research, numerous plant species have been studied which can be used to remediate the HM polluted soils, including *Brassica napus* for Pb [9], cadmium (Cd) [10], and copper (Cu) [11], while cauliflower and sunflower were used to remediate chromium (Cr) [12,13], accordingly.

Usually, many of the HMs are adsorbed in soil particles to make soil aggregates that are hard to be integrated by plants. Thus, the use of acids, which are low molecular weight organic acids like citric acid (CA), is crucial to alter the chemical activity/bioavailability of HMs and improve phytoextraction [14]. Although phytoextraction is manageable, cost-effective, and eco-friendly, it is not recommended commercially. The capacity of Pb-phytoextraction can be effectively improved by the selection of effective chelating agents like CA. Recent studies have documented the role of CA as a growth promoting agent with a chelating potential against different HMs such as Cr [15], copper (Cu) [11], Pb [16] as well as Cd [10].

Castor beans (*Ricinus communis* L.) is a tropical African plant that is well-known for its ability to grow in contaminated sites, which makes this plant a promising candidate to clean up the environment and re-cultivate polluted lands [17]. Previous scientific research has suggested that castor beans grown in specific areas of the world can accumulate Pb [18]. Castor beans is an industrial crop, is used other than in food items, and has tremendous potential in a rotation system. Therefore, it is a novel crop to cultivate for the phytoremediation of HMs because this crop has an additional economic benefit if grown in polluted sites [19].

Most of the plants used for the phytoextraction of Pb contaminated soils belong to the temperate climate, and less is known about the ability of tropical plants for the phytoextraction of HMs. By keeping in view the significance of phytoextraction to remediate polluted lands, the quest for tropical plants becomes obligatory. Therefore, this study analyzes the Pb-induced harmful effects on morph-physiological and biochemical attributes of castor beans. In addition, Pb uptake and its detoxification by the application of CA are also explored, which can be used to remediate Pb contaminated sites.

## 2. Results

### 2.1. Plant Growth and Biomass

Figure 1A–F demonstrates the negative changes in growth and biomass produced due to Pb-induced toxicity as well as the positive effect of CA application on growth and biomass characteristics of the plant. Lead application at both levels (300 and 600 mg kg^−1^) substantially reduced plant (root and shoot) length, leaf area, the number of leaves per plant, and dry weight, and the response was dose-dependent for all parameters (Figure 1A,B). In contrast to the control treatment, the decrease in the shoot and root length was 7.21% (300 mg kg^−1^) and 18.43% (600 mg kg^−1^), respectively, under low and high levels of Pb stress while both doses of CA potentially improved the root and shoot lengths of Pb-exposed castor beans plants. Similarly, shoot and root dry weight, the number of leaves per plant of castor beans decreased significantly when exposed to Pb stress. Application of CA (2.5 and 5 mM) considerably improved plant growth and biomass parameters under Pb stress compared to Pb-stress-alone, as demonstrated in Figure 1A–F.

### 2.2. Photosynthetic Pigments and Gas Exchange Parameters

Exposure to Pb stress at both levels (300 and 600 mg kg^−1^) slightly decreased (*p* < 0.05) the photosynthetic pigments including chlorophylls (a and b), carotenoids, and total chlorophyll compared to the relevant control treatments (Figure 2A–D). When CA was applied to Pb-stressed plants, a significant improvement in all the photosynthetic parameters was observed compared to control and Pb-alone treatments (Figure 2A–D). Moreover, CA also improved these parameters where no Pb stress was applied. Citric acid (5 mM) increased the chlorophylls a and b, total chlorophyll, and carotenoids by 35, 42, 37, and 42%, respectively. Similarly, photosynthetic characteristics, including photosynthetic rate, water use efficiency, stomatal conductance, and transpiration rate, decreased significantly in Pb-exposed plants without CA (Figure 3A–D). The citric acid application significantly improved all the gas exchange characteristics, including photosynthetic rate, transpiration rate, water use efficiency, and stomatal conductance up to 33%, 37%, 43%, 47%, respectively, in Pb-exposed castor beans plants with respect to the Pb-alone treated plants (control).

### 2.3. Antioxidant Enzymes Activities

The effects of Pb stress (300 and 600 mg kg^−1^) and CA (2.5 and 5 mM) application (alone or combined) on important antioxidant enzyme activities, i.e., POD, CAT, APX, and SOD, in roots and leaves of castor beans plants are shown in Figure 4A–H. A significant reduction in the activities of antioxidant enzymes was observed in Pb-alone treatments over the control. Although higher Pb stress (600 mg kg^−1^) showed more reduction in enzyme activities than lower Pb stress (300 mg kg^−1^), this decrease was significant (*p* < 0.05) in all the plants expose to Pb toxicity as compared to control plants. Under both Pb-stress levels, CA treatments significantly (*p* < 0.05) enhanced the efficiency of these enzymes compared to their respective Pb-alone treated roots and leaves of the plants.

### 2.4. Effect of CA Application on EL, H_2_O_2_, and MDA 

With exposure to Pb stress, electrolyte leakage (EL), H_2_O_2,_ and MDA concentrations were significantly enhanced in castor beans plants (leaves and roots) (Figure 5A–F). Application of CA at both levels (2.5 and 5 mM) to castor beans plants significantly (*p* < 0.05) recovered the oxidative damage caused by Pb, and the positive effects of CA were more prominent in the 5 mM CA treatment than the lower CA level. With the 5 mM CA + 600 mg kg^−1^ Pb application, the EL, H_2_O_2_, and MDA contents in leaves significantly (*p* < 0.05) reduced by 32%, 18%, and 21%, respectively, as compared with the respective control without CA, while in roots this reduction was 18%, 20%, and 33%, respectively.

### 2.5. Lead Concentration and Uptake in Castor Beans 

The concentrations of Pb in root and shoots of castor beans were significantly enhanced with the increase in doses of Pb (Figure 6A,B). The castor beans plants stored higher Pb concentration in roots compared to shoot tissues, regardless of the applied Pb levels. The application of CA at a higher dose (5 mM) significantly increased the Pb concentration in the shoots and roots of castor beans plants. This increase in Pb concentrations was about 26% and 18% in shoots, and 25% and 28% in roots at 300 and 600 mg kg^−1^ Pb concentrations, respectively, compared to Pb-alone treated castor beans plants. Similarly, CA application at the higher dose (5 mM) significantly increased the total Pb uptake in castor beans plants; for example, the increase in total shoot Pb uptake was 46% and 39%, while in roots this increase was 48% and 49% at lower and higher levels of the Pb application, respectively (Figure 6C,D).

## 3. Discussion

### 3.1. Plant Growth and Biomass

This study was conducted for the determination of potential effects of Pb stress on castor beans plants, the potential of castor beans plants toward the phytoremediation of Pb, and CA’s role towards the phyto-extraction of Pb by this plant. The effect of Pb on morphological attributes is prominent in our results (Figure 1), which clearly showed the retarded growth of plants under Pb stress alone. The high concentration of Pb decreased the roots and shoot lengths, plant height, and fresh and dry biomass. The greater uptake of the Pb in the plant can restrict the biomass production of plants [20,21]. The higher Pb toxicity in plants caused growth inhibition, reduced the number of leaves, and produced smaller, more brittle leaves in maize plants [22]. These detrimental effects of Pb on plant growth might be the result of a disturbance in the nutrient metabolic process in plants [23]. Lead, even at low concentrations, hampers the growth of the aerial parts and roots of plants [24], which also support the results of our study. Pb exposure considerably decreased root expansion in *Prosopis* sp. [25]. Similarly, Pb exposure in *A. Sativum* badly damaged the mitochondria, vacuolization of endoplasmic reticulum, and plasma membrane in the roots [26]. Malar et al. [27] also reported similar results, as observed in our study, and reported reduced plant growth and biomass in *H. incana* under Pb stress. Similarly, Fahr et al. [28] described the reduction in root length, plant height, fresh and dry biomass in Pb-stressed *Brassica* plants. This reduction in morphological attributes of plants might be due to the fact that Pb toxicity disturbs the intracellular biochemical reaction, cell division, and DNA synthesis in the same way as described in other various studies under HM stress [29,30,31].

The promotive role of CA in plants exposed to HM stress is well recognized [11,32]. The application of CA improved the growth and biomass of castor beans plants under Pb stress (Figure 1). Results of the present study were in line with the outcome described by Ehsan et al. [10], Farid et al. [13], and Afshan et al. [15] for various plants species. These studies have demonstrated that the CA application improved plant growth under various metal stresses. The application of CA plays a vital role under metal stress and promotes plant growth and biomass, which may be because of an increase in nutrient uptake by plants [33]. Improvement in biomass and plant growth might be accredited to the ability of CA to enhance the uptake of essential nutrients by the formation of complexes with nutrients [34]. The other possible reason might be that the application of CA may enhance the photosynthesis and synthesis of phytochelatins (PCs) in plants [33].

### 3.2. Photosynthetic Pigments

The photosynthetic pigments played a vital role in plant life as they harvest light for biochemical reactions in leaf cells. For an HM tolerance ability of a plant, chlorophyll contents may be considered one of the important indicators of HM stress [35]. Our results indicated that Pb stress significantly reduced the photosynthetic pigment in plants, which ultimately lead to the diminution in plant growth and biomass (Figure 2). A recent study had been conducted where Pb significantly affected the photosynthetic pigments in different plants [16,36]. This reduction in photosynthetic pigments might be due to the result of Pb stress, which may affect the structure of chloroplast, chloroplast membranes, plastoglobuli, promote the ion exchange in the chloroplast and inhibit the essential enzymes in the Calvin cycle [30]. All of these phenomena lead to the reduction of photosynthetic pigments, which ultimately reduce the growth of plants. The application of CA enhanced the photosynthetic pigment in stressed plants (Figure 2). A useful role of the CA towards the photosynthetic system in plants, when exposed to HMs stress, has been described in many recent research works [13,15,16]. This might be the effect of CA application, which enhanced the uptake of essential nutrients as well as the formation of photosynthetic pigments [13]. Enhancement in photosynthetic pigments in photochemical reactions may transform the light energy more effectually under the application of CA, resulting in enhancement of biomass and plant growth [30]. Citric acid significantly enhanced the contents of photosynthetic pigments of *B. napus* under Cu stress [11] and of *I. lacteal* under Pb stress [37], as consistent with our results. Similarly, the application of CA mitigated the adverse effect of Pb in *S. durmmondii* and enhanced photosynthetic activity as well as plant growth [38]. 

### 3.3. Antioxidant Enzymes

Plants develop their defensive system in the template of the antioxidant enzyme system which has a vital part in the alleviation of oxidative stress. In the current study, the activities of CAT, SOD, and POD were considerably reduced under Pb stress (Figure 3). In numerous studies, it has been reported that Pb stress reduced the activities of antioxidant enzymes in various plants [10,13,39]. Sometimes antioxidant enzymes exhibit dual behavior: mild HM stress increases the antioxidant enzyme activities while higher HM stress reduces the enzymatic activities of antioxidant enzymes. Antioxidant enzymes (SOD, POD, and CAT) activities increased in sunflower with an increase in the concentration of Cr in the soil [13]. In another study, the same response of antioxidant enzymes was observed in wheat, *B. napus*, and mung bean at a lower concentration of Cr toxicity [40]. Increased levels of CA (2.5 and 5 mM) enhanced enzymatic activities at all concentrations of Pb regarding the toxicity of Pb (Figure 3). This might be due to the growth promotive character of CA in assisting the plant to recover fast from oxidative damage [32]. In addition, the ameliorating effect of CA also helps the plant to alleviate Pb stress and to increase the activities of antioxidant enzymes, as investigated in sunflower plants [41]. Increased levels of EL and ROS also frequently increase the anti-oxidative enzyme activities as the plant tries to alleviate the metal-induced stress by activating the enzymatic defense system [42]. Citric acid application in cotton improved the antioxidant enzyme activities by mitigating Ni stress [11]. Similar kinds of improvements in antioxidant enzymes through the exogenous application of CA (2.5 and 5 mM) have been observed in *B. napus* under different metal stress [10,15,16]. 

### 3.4. H_2_O_2_, EL, and MDA Contents

The H_2_O_2_, EL, and MDA contents in plant cells indicate the degree of oxidative stress and peroxidation of membranes. Plants exposed to Pb stress confronted oxidative stress by the EL and production of ROS [12]. In this study, H_2_O_2_ and MDA contents and EL increased markedly with an increasing concentration of Pb in soil (Figure 5). Similar outcomes have been described in recent studies performed on wheat, barley, and *B. napus* in which the toxicity of Cr resulted in oxidative stress and a higher level of EL [40]. Similarly, sunflower showed a high production of MDA and H_2_O_2_ under HM stress [43]. It has already been reported that Pb stress enhanced the production of ROS, which induced severe oxidative stress in plant cells [13,39]. The ability of HMs to damage the electron transport chain and K^+^ efflux may enhance the production of O^-^ and OH^-^ radicals and, ultimately, EL in plants [44]. The biotic and abiotic stress fortified by metals also increase the EL in plants [45,46]. The oxidative damage is directly correlated with a reduction of growth in plants as well as biomass [47]. In the current study, the application of CA predominantly reduced the generation of EL and ROS in castor beans plants by facilitating the activities of antioxidant enzymes. Antioxidant defensive activities play a novel role regarding the reduction of EL and ROS production [13,48]. A similar impact of CA was also detected in *B. napus* and sunflower under the stress of Cr toxicity [13,15]. In this study, the increase in castor beans growth and biomass by application of CA may be attributed to the inhibiting effect of CA on EL and ROS production. 

### 3.5. The Accumulation of Pb in Roots and Leaves

Concentrations of Pb in roots and shoots of castor beans plants exposed to different levels of Pb along with applications of CA are shown in Figure 6. Our results clearly explained that Pb uptake increased with increasing concentrations of applied Pb in the system of soil. It is well observed that the accumulation of Pb in different parts of the plant correlates with the concentration of Pb in growth media [47]. Our results clearly showed that the concentration of Pb were significantly higher in roots as compared with the aerial parts of the plant. Similar results have been reported by many researchers [10,13,47]. That might be due to the ability of Pb to make complexes with polysaccharides such as sugar and other macromolecules.

The application of CA enhanced the uptake of Pb in roots and shoots of the castor bean. It was reported that CA application enhanced both uptake and translocation of various HMs in plants [32,49]. The chelating role of CA in soil enhances the movement and availability of HMs (Cd, Pb, Cu, and Zn) in the soil. The chelating effect of CA also made the complex with metals and enhanced the accumulation of metals by plants [50]. For example, the application of CA enhanced the uptake of Cr in sunflower [13,41]. Our results related to Pb concentrations in tissues are comparable with Kiran and Prasad [51] in which rice husk ash and biochar assisted the phytoremediation potential of castor beans plants for Pb-spiked soil. However, in our study, the Pb uptake was slighter higher as compared with Kiran and Prasad [51]. The higher Pb uptake by castor beans plants in our study might be due to the difference in plant variety, soil texture, experimental duration, and varying experimental conditions. Moreover, we applied Pb in soluble form which can be easily taken up by plants. The other potential cause of higher Pb concentrations in plants might be that the CA may decline the pH of the soil, which upgrades the mobility of metals and finally improves the uptake of HMs by plants [52,53]. Citric acid application also increased plant growth and biomass and, consequently, the accumulation and uptake of metals in plants [13,54]. Pb accumulation was found both in roots and shoots whereas maximum accumulation was found in the roots of castor beans plants. It is reported that plants have more capacity to accumulate metals in roots than in shoots. The accumulation of Pb in roots has limited the transfer of HMs to above-ground tissue as reported earlier [55]. The finding of our study showed that the application of CA improved the phytoextraction of Pb via castor beans plants and increased the biomass and growth of the plants, which might be employed for the cultivation of metal contaminated soils. 

## 4. Materials and Methods

### 4.1. Soil Sampling and Analysis

Soil sampling was made from an agricultural field of the University of Agriculture Faisalabad, Pakistan. Roots and other large debris were removed from the soil and sieved by 2 mm sieve. Before conducting the experiment, basic soil analysis was performed to check the soil properties. Standard protocols were followed for various soil analysis, i.e., pH and EC by Soltanpour [56], sodium adsorption ratio and soluble ions by US Salinity Laboratory Staff [57], and Page et al. [58]. Soil particle size was determined through a protocol described by Bouyoucos [59]. The Walkley–Black method was followed for soil organic matter determination described by Jackson [60] and the calcimeter method was followed for calcium carbonate [61]. Detailed soil properties are described in Table 1. 

### 4.2. Pot Experiment

A pot experiment was conducted, and for this purpose, the plastic pots, with 5 kg capacity, were filled with sieved soil. Five castor beans seeds were sown in each pot. Right after germination, plant thinning was made, and 2 plants were maintained in each pot. A complete randomized design was applied along with 3 replicates against each treatment. Plants were fertilized using NPK with a ratio of 120:50:25 kg ha^−1^. For NPK, the salts of urea, diammonium phosphate, and potassium sulfate were used, respectively. Various concentrations of Pb (0, 300, and 600 mg kg^−1^) were applied using PbNO_3_ salt. First time, Pb treatments were applied after 30 days of sowing. Then, the remaining Pb solution was applied after a 7-day interval with four times of application. The total calculated amount of Pb for each treatment and replicates was added in 2 L of water, and 500 mL of water was applied each time. The final Pb concentrations in the soil were either 300 or 600 mg kg^−1^ after the four-times of soil application. This practice of Pb soil application was performed to make sure of Pb in soluble form in the soils. The higher level of Pb was selected for the identification of mechanisms of CA on the reduction of Pb toxicity in plants. Nitrogen added through the Pb salt was calculated and adjusted through N fertilizers. This was performed to avoid the effect of N in the study. The experimental plants were provided with a foliar spray of different concentrations of CA (0, 2.5, and 5 mM) at the time of Pb application in the soil, and it was provided in different intervals, similar to the Pb application. The total volume of CA used was 1 L for each treatment for all replicates. 

### 4.3. Plants Harvesting

After 70 days of sowing, the harvesting of castor beans plants was made and their different parts were differentiated. After washing with distilled water, plants were dried in oven (70 °C) for 72 h and dry weights were recorded. Diluted HCl acid (1.0%) was used for root washing; after that roots were washed with distilled water several times until the acidic content had been fully removed. After drying at room temperature, roots were also oven-dried at 70 °C, and dry weight was noted. 

### 4.4. Chlorophyll Contents and Gas Exchange Parameters

For estimation of chlorophyll contents, the fresh leaf samples were collected and centrifuged after the extraction in 85% acetone (*v/v*), and readings were taken at recommended wavelengths with a spectrophotometer [62]. On the same day in full sunshine (10:00–12:00 a.m.), IRGA was used for readings of photosynthetic rate, transpiration rate, stomata conductance, and water use efficiency. 

### 4.5. Estimation of MDA, EL, H_2_O_2_ and Antioxidants Enzymes

The Zhang and Kirkham [63] method was followed to measure MDA contents by using thiobarbituric acid (0.1%), further described by Abbas et al. [64]. The Dionisio-Sese and Tobita [65] method was used for EL estimation. For this, the initial and final EC of the solution was noted through the extraction of samples for 2 h at 32 °C and then doing the extraction of the same samples at 121 °C for 20 min, respectively. To measure the contents of H_2_O_2_, the Jana and Choudhuri [66] procedure was followed. In brief, samples were homogenized with phosphate buffer (50 mM) at 6.5 pH and centrifuged for 20 min. Then H_2_SO_4_ (20%, *v/v*) was added and centrifuged again for 15 min, and absorbance was noted at 410 nm.

Samples were crushed in liquid nitrogen and standardized in 0.5 M phosphate buffer at a pH of 7.8 for the investigation of SOD and POD activities [67]. APX and CAT activities were estimated by the following procedures of Nakano and Asada [68] and Aebi [69], respectively. The detailed procedures are described by Abbas et al. [63]. 

### 4.6. Estimation of Pb Contents 

The 1.0 g of each sample was digested at a hot plate with 4:1 of HNO_3_:HClO_4_ (*v/v*), and the concentration of Pb was estimated in digested samples by an atomic absorption spectrophotometer [70].

### 4.7. Statistical Analysis

At 5% probability level, ANOVA was applied for data analysis by using SPSS software (Statistics Software, Version 21.0). Tukey’s HSD post hoc test was applied for multiple comparisons of the means.

## 5. Conclusions

The current study has shown that the promising application of CA remarkably alleviates Pb toxicity in castor beans plants at various morpho-physiological and biochemical levels. Application of CA significantly improved the content of Pb in castor beans plants and increased the growth of the plants as well as the antioxidant defense system, which further supported the plants’ normal functioning and metabolism under Pb stress.

## Figures and Tables

**Figure 1 plants-08-00525-f001:**
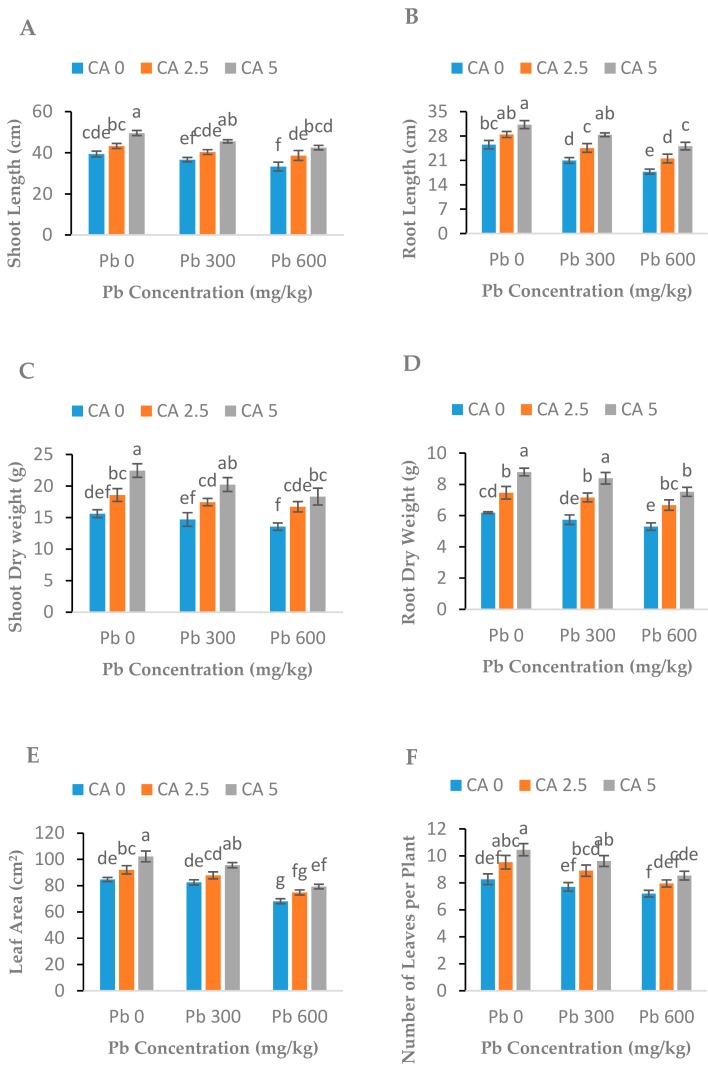
Effect of different Pb concentrations (0, 300, and 600 mg kg^−1^) and CA levels (0, 2.5, and 5 mM) on shoot lengths (**A**), root lengths (**B**), shoot dry weight (**C**), roots dry weight (**D**), leaf area (**E**), numbers of leaves per plant (**F**) of castor beans plants. Values reported in figures are the mean of 3 replicates along with standard deviation. Letters show the significant difference between the treatments of *p* ≤ 0.05.

**Figure 2 plants-08-00525-f002:**
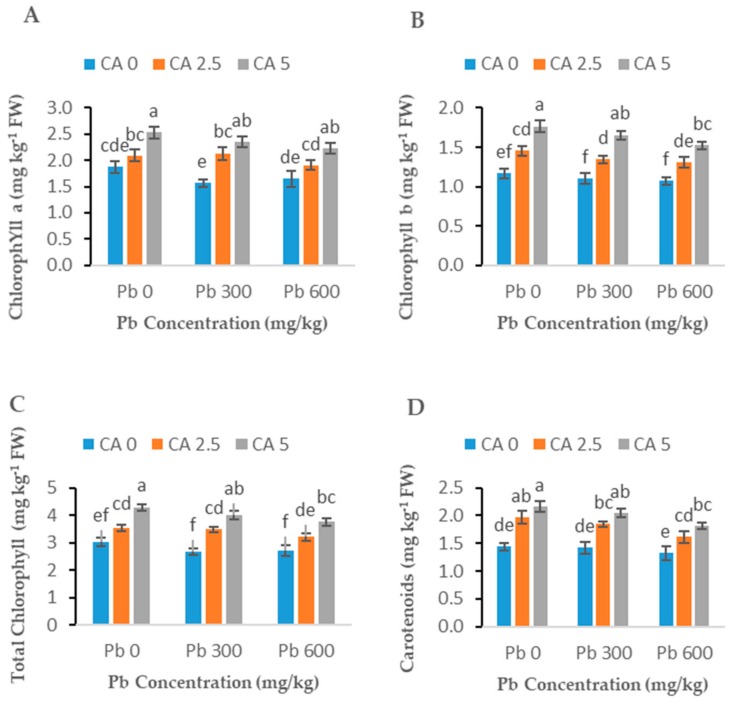
Effect of different Pb concentrations (0, 300, and 600 mg kg^−1^) and CA levels (0, 2.5, and 5 mM) upon chlorophyll *a* content (**A**), chlorophyll *b* content (**B**), chlorophyll total (**C**), and the carotenoids content (**D**) of castor beans plants. Values reported in figures are the mean of 3 replicates along with standard-deviation. Letters show the significant difference between the treatments of *p* ≤ 0.05.

**Figure 3 plants-08-00525-f003:**
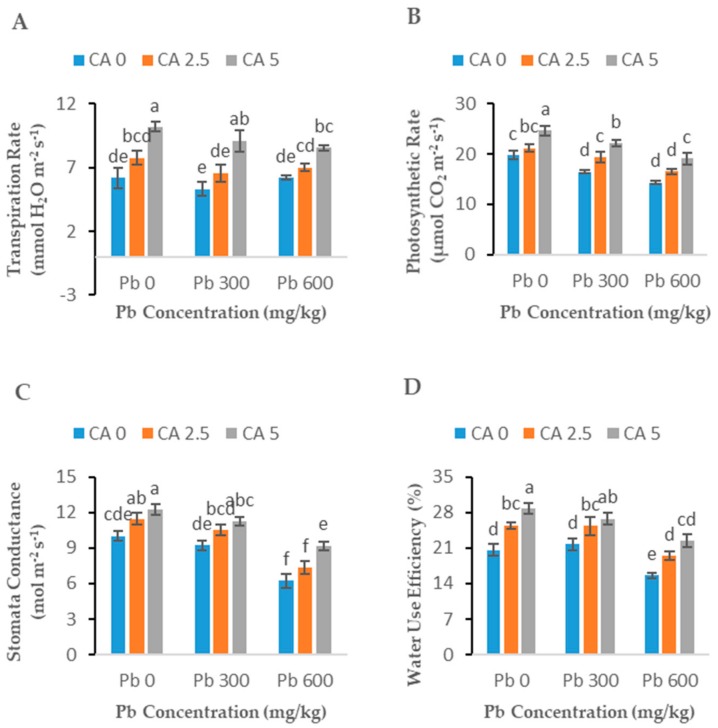
Effect of different Pb concentrations (0, 300, and 600 mg kg^−1^) and CA levels (0, 2.5, and 5 mM) on transpiration rate (**A**), photosynthetic rate (**B**), stomata conductance (**C**), water use efficiency (**D**) of castor beans plants. Values reported in figures are the mean of 3 replicates along with standard deviation. Letters shows the significant difference between the treatments of *p* ≤ 0.05.

**Figure 4 plants-08-00525-f004:**
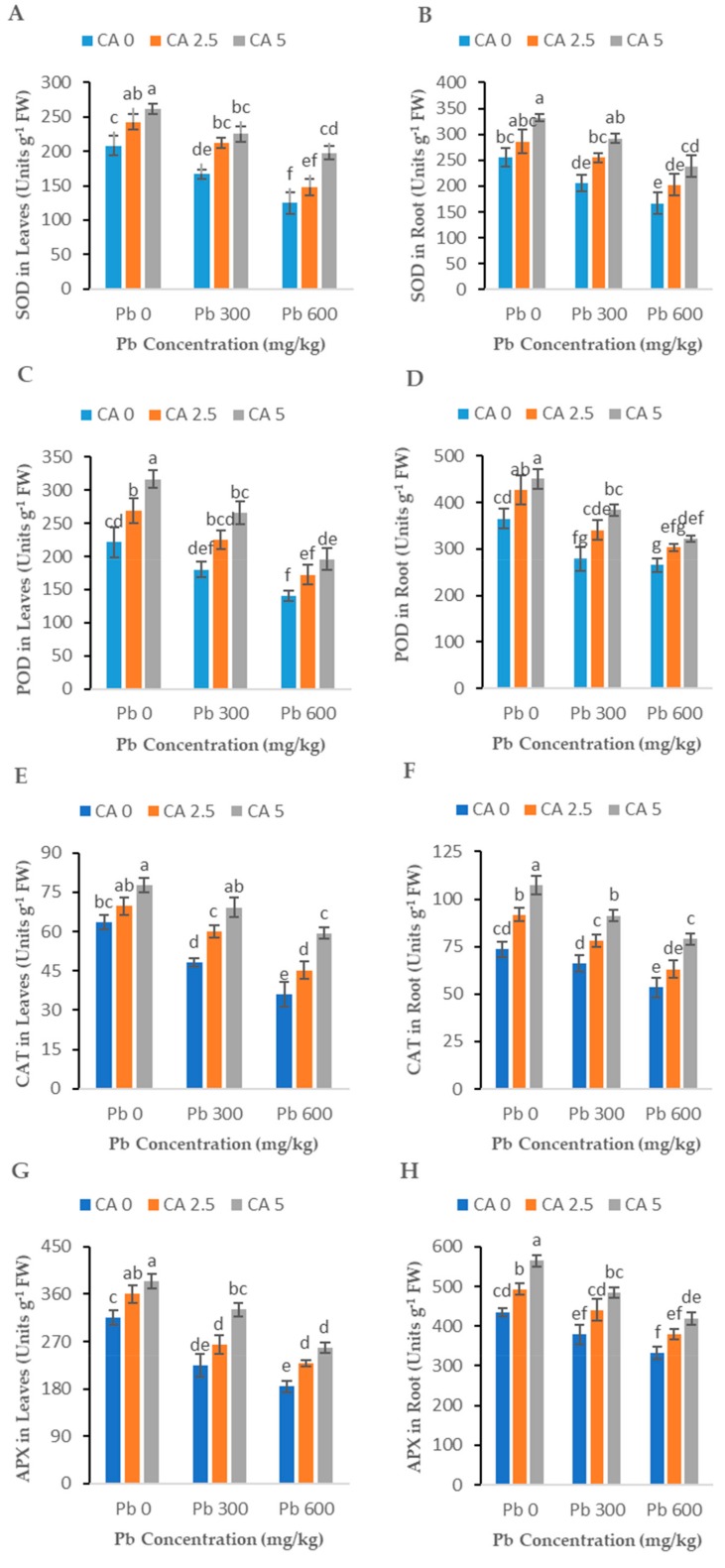
Effect of different Pb concentrations (0, 300, and 600 mg kg^−1^) and CA levels (0, 2.5, and 5 mM) on leaf SOD content (**A**), roots SOD content (**B**), leaf POD content (**C**), roots POD content (**D**), leaf CAT content (**E**), roots CAT content (**F**), leaf APX content (**G**), roots APX content (**H**) of castor beans plants. Values reported in figures are the mean of 3 replicates along with standard deviation. Letters show the significant difference between the treatments of *p* ≤ 0.05.

**Figure 5 plants-08-00525-f005:**
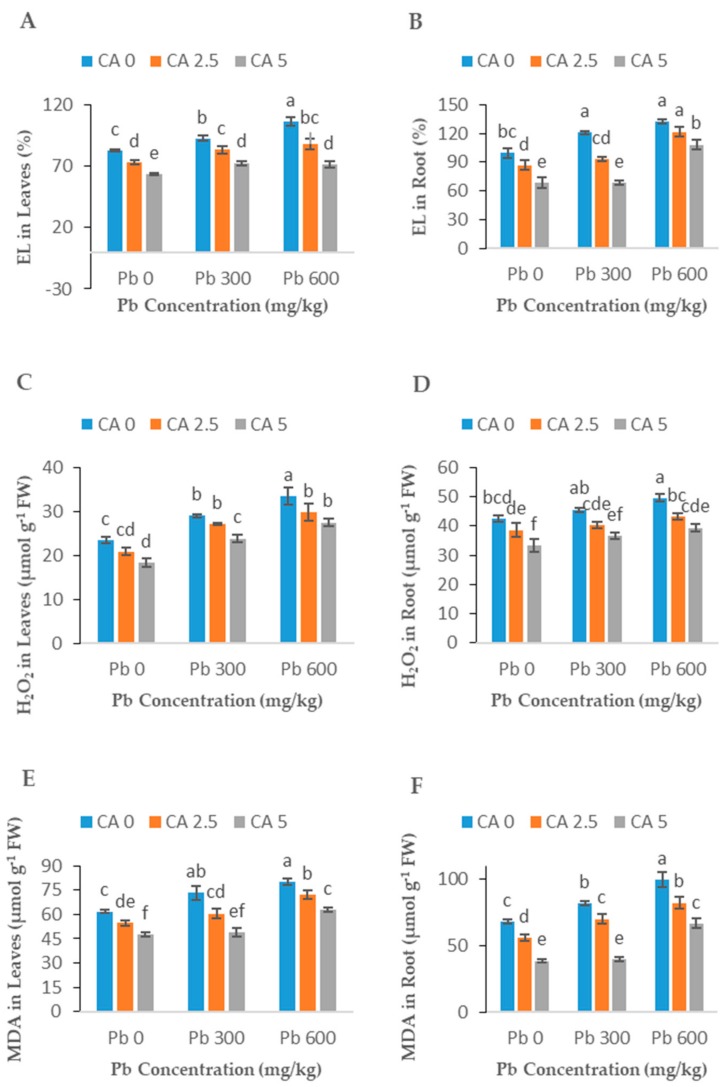
Effect of different Pb concentrations (0, 300, and 600 mg kg^−1^) and CA levels (0, 2.5, and 5 mM) on EL in leaf (**A**), EL in roots (**B**), leaf H_2_O_2_ content (**C**), roots H_2_O_2_ content (**D**), leaf MDA content (**E**), roots MDA content (**F**) of castor beans plants. Values reported in figures are the mean of 3 replicates along with standard deviation. Letters show the significant difference between the treatments on *p* ≤ 0.05.

**Figure 6 plants-08-00525-f006:**
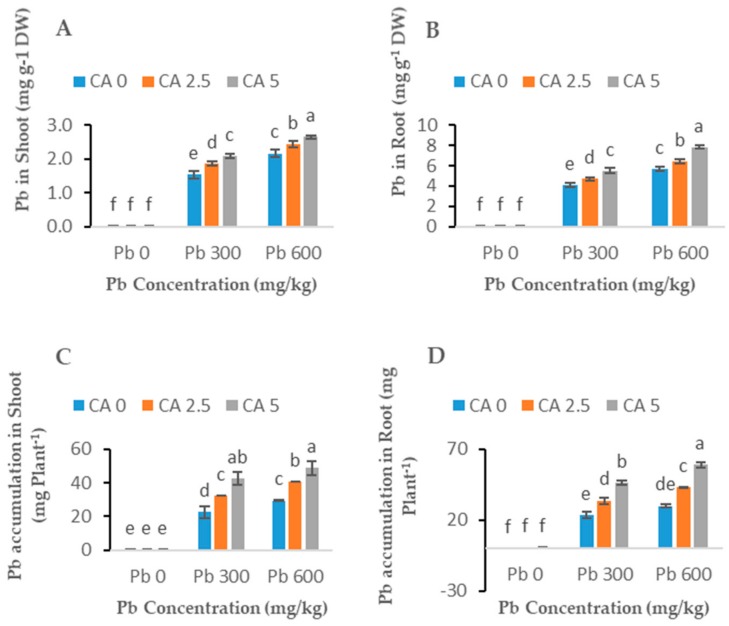
Effect of different Pb concentrations (0, 300, and 600 mg kg^−1^) and CA levels (0, 2.5, and 5 mM) on Pb concentration to the plant shoot (**A**), Pb concentration to the plant root (**B**), Pb accumulation in shoot (**C**), Pb accumulation in root (**D**) of castor beans plants. Values reported in figures are the mean of 3 replicates along with standard deviation. Letters show the significant difference between the treatments of *p* ≤ 0.05.

**Table 1 plants-08-00525-t001:** Soil physico-chemical properties used for the experiment.

Texture	Sandy Loam
Silt	15.0%
Sand	67.9%
Clay	17.1%
EC	1.96 dS m^−1^
pH	7.61
Sodium adsorption ratio (SAR)	1.89 (mmol L^−1^)^1/2^
Available P	2.11 mg kg^−1^
Organic matter	0.59%
HCO_3_	2.51 mmol L^−1^
SO_4_^−2^	11.44 mmol L^−1^
Cl^-^	5.45 mmol L^−1^
Ca^2+^ + Mg^2+^	13.98 mmol L^−1^
K^+^	0.04 mmol L^−1^
Na^2^	5.23 mmol L^−1^
Available Zn^2^	0.77 mg kg^−1^
Available Cu^2+^	0.31 mg kg^−1^
Available Cr ^+6^	0.16 mg kg^−1^

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
