# Peer review of "Citric Acid Enhances Plant Growth, Photosynthesis, and Phytoextraction of Lead by Alleviating the Oxidative Stress in Castor Beans"

_plants, 2019, doi:10.3390/plants8110525_

Round 1

Reviewer 1 Report

Review of Mallhi et al. Citric Acid Enhances Plant Growth….         

The paper reports the effects of citric acid addition to soils spiked with lead on various plant growth parameters. The results show a beneficial effect of citric acid on plant growth in the presence of lead. On the surface, the paper is straightforward, nothing extraordinary or novel. The effect of citric acid and other chelants on lead uptake has been well documented as well as the beneficial effects on plant growth of soil applied chelants and organic acids under certain conditions. There are some things that don’t make sense to me and cause me to wonder if we are getting the full picture. Additionally, there are some explanations by the authors that do not convey an understanding of lead chemistry in the rhizosphere and root surface. The use of spiked soils, without allowing equilibration or “aging”, the lack of description of how the soils were spiked and how citric acid was applied result in serious questions about the results reported.

The authors show data that castor bean plants growing in the presence of 300 or 600 mg Pb/kg soil accumulate in excess of 1.5 mg Pb/g (1500 mg/kg) plant matter in the absence of citric acid. That result in itself is astounding, yet the authors hardly mention it. Is that correct? That would be extremely surprising. In the absence of chelants, soil grown (or even hydroponically grown) plants rarely accumulate Pb concentrations in above ground tissues more than 10-50 mg/kg or possibly up to 100 mg/kg. On occasion researchers have observed high concentrations in plants but it is usually the result of unrealistic hydroponic conditions or where extreme phytotoxicity occurs and root membranes become leaky. The solubility of lead in the conditions found in the root including the presence of phosphate, free carbonate, iron and neutral to mildly acidic pH does not typically support translocation of lead to the above ground plant tissue. Numerous studies have used advanced imaging to show the fate of Pb in plant roots. The addition of chelants to the growth media can complex the lead and prevent precipitation and sorption to allow translocation, but in the absence of those chelants it is extremely rare to see these levels of translocation. This surprising accumulation is also confounded by the addition of NPK fertilizers to the soil. Lead phosphate forms readily under the soil conditions described (alkaline pH, added phosphate fertilizers) and its solubility product would not allow dissolution by addition of citrate. In my experience this rate of accumulation in the control plants in the absence of citrate is very surprising and difficult to believe – if this actually occurred as described, this result is substantially more important than any of the effects reported for citric acid. Phytoextraction of Pb is limited because the only way to obtain any substantial levels of accumulation is through the use of soil applied chelants, yet the authors seem completely oblivious of the significance of the magnitude of Pb reported in the plant shoots with only a minimal effect on plant biomass yield compared to the 0 lead treatments.

In this study soils were spiked with Pb, no information is presented on the form of Pb added to the soil, but one would have to assume based on the results that the Pb was maintained in a soluble form. This presents some difficulty in extrapolating these results to actual conditions where Pb solubility is very low. It is extremely rare to see phytotoxicity from lead even with total Pb concentrations in the soil exceed 5000 mg/kg. Phytotoxicity from lead in natural field conditions occurs rarely and easily ameliorated through the addition of phosphate. As a result I don’t see a great deal of utility or application for the research described here. That does not negate the results, just difficult to see any importance.

The effect of citrate on increasing lead uptake has been reported previously. In this paper because the lead uptake in the control (no added citrate) was so high, the citrate only provided a modest increase in lead uptake and yield. Under field conditions, where Pb uptake and solubility in the soil is much lower this effect would likely be insignificant.

2 line 92 (and p 10 line 294) – The application of citric acid needs to be explained more accurately and thoroughly. This is an integral component of the work and the methods used will affect the results. The application of 2.5 mM or 5 mM Citric acid only indicates the concentration of citric acid applied – what quantity of solution was applied to the soil? Was the solution irrigated into the soil, was any leachate retained in the pot and reabsorbed by the soil or was it free draining (did any citric acid leave the system)? The concentrations (2.5 mM and 5 mM) should be converted to a rate, i.e., mmol/kg soil or mmol/cm2 surface area. Applying 50 ml of 2.5 mM CA would not be the same as applying 500 mL of CA. Was Citric Acid applied as the acid form or as a sodium salt (sodium citrate).

P 2 line 72 – the cited references discusses Cd uptake in Castor bean – is there an original (better?) reference for Pb uptake in shoots of Castor bean?

P 10 line 296 – “At maturity” – how are you defining maturity – is it flowering, seed set or senescence – how long were the plants grown prior to harvest.

10 line 288 – how was the soils spiked – what form of lead was used. Addition of a soluble Pb salt such as lead nitrate unless allowed to equilibrate and “age” yields unnaturally high levels of soluble Pb. Additionally, as the soluble Pb hydrolyzes the soil pH will become more acidic resulting in substantially different chemical characteristics than what is presented in Table 1. I believe it very important that the soil characterization data should be presented for soil after spiking, including the actual measured concentrations of Pb in the soil after spiking – likewise the control (0 mg/kg Pb) should be spiked with a similar soluble salt so that nitrate, for example if lead nitrate was used, are at similar levels in the soil. 9, line 260 – I believe it to be widely accepted that Pb is retained on the root surface through precipitation as lead oxides, hydroxides, carbonates and phosphates, not complexed with sugars.

Table 1 – soil characterization data presents some of the components as mmol/L – I assume this must be the concentration in the extract (saturated paste, 1:1, etc.)? – Those values should be normalized and presented on a dry soil basis (mg/kg soil, dry weight) as soluble anions

Author Response

Response to Reviewers’ comments

Title: Citric Acid Enhances Plant Growth, Photosynthesis and Phytoextraction of Lead (Pb) by Alleviating the Oxidative Stress in Castorbean

Reviewer 1

The paper reports the effects of citric acid addition to soils spiked with lead on various plant growth parameters. The results show a beneficial effect of citric acid on plant growth in the presence of lead. On the surface, the paper is straightforward, nothing extraordinary or novel. The effect of citric acid and other chelants on lead uptake has been well documented as well as the beneficial effects on plant growth of soil applied chelants and organic acids under certain conditions. There are some things that don’t make sense to me and cause me to wonder if we are getting the full picture. Additionally, there are some explanations by the authors that do not convey an understanding of lead chemistry in the rhizosphere and root surface. The use of spiked soils, without allowing equilibration or “aging”, the lack of description of how the soils were spiked and how citric acid was applied result in serious questions about the results reported.

Response

Thank you very much for giving us a chance to respond to the comments. We have made every effort to address each reviewer’s concerns as clearly and succinctly as possible, as demonstrated in the following pages

Thank you once again for this opportunity and please let us know if you have any questions or concerns regarding the paper.

Sincerely,

Dr. Shafaqat Ali

The authors show data that castor bean plants growing in the presence of 300 or 600 mg Pb/kg soil accumulate in excess of 1.5 mg Pb/g (1500 mg/kg) plant matter in the absence of citric acid. That result in itself is astounding, yet the authors hardly mention it. Is that correct? That would be extremely surprising. In the absence of chelants, soil grown (or even hydroponically grown) plants rarely accumulate Pb concentrations in above ground tissues more than 10-50 mg/kg or possibly up to 100 mg/kg. On occasion researchers have observed high concentrations in plants but it is usually the result of unrealistic hydroponic conditions or where extreme phytotoxicity occurs and root membranes become leaky. The solubility of lead in the conditions found in the root including the presence of phosphate, free carbonate, iron and neutral to mildly acidic pH does not typically support translocation of lead to the above ground plant tissue. Numerous studies have used advanced imaging to show the fate of Pb in plant roots. The addition of chelants to the growth media can complex the lead and prevent precipitation and sorption to allow translocation, but in the absence of those chelants it is extremely rare to see these levels of translocation. This surprising accumulation is also confounded by the addition of NPK fertilizers to the soil. Lead phosphate forms readily under the soil conditions described (alkaline pH, added phosphate fertilizers) and its solubility product would not allow dissolution by addition of citrate. In my experience this rate of accumulation in the control plants in the absence of citrate is very surprising and difficult to believe – if this actually occurred as described, this result is substantially more important than any of the effects reported for citric acid. Phytoextraction of Pb is limited because the only way to obtain any substantial levels of accumulation is through the use of soil applied chelants, yet the authors seem completely oblivious of the significance of the magnitude of Pb reported in the plant shoots with only a minimal effect on plant biomass yield compared to the 0 lead treatments.

In this study soils were spiked with Pb, no information is presented on the form of Pb added to the soil, but one would have to assume based on the results that the Pb was maintained in a soluble form. This presents some difficulty in extrapolating these results to actual conditions where Pb solubility is very low. It is extremely rare to see phytotoxicity from lead even with total Pb concentrations in the soil exceed 5000 mg/kg. Phytotoxicity from lead in natural field conditions occurs rarely and easily ameliorated through the addition of phosphate. As a result I don’t see a great deal of utility or application for the research described here. That does not negate the results, just difficult to see any importance.

The effect of citrate on increasing lead uptake has been reported previously. In this paper because the lead uptake in the control (no added citrate) was so high, the citrate only provided a modest increase in lead uptake and yield. Under field conditions, where Pb uptake and solubility in the soil is much lower this effect would likely be insignificant.

Response

Thank you very much your valuable comments. Actually, the term “soil-spiking” was typing error. We have applied the Pb concentrations in soluble form, not in soil spiking. Our results are comparable with recently published study (accepted 14 Aug 2019; available online 20 Aug 2019) i.e. “Kiran, B.R; and Prasad, M.N.V. Biochar and rice husk assisted phytoremediation potentials of Ricinus communis L. for Lead-spiked soils. Ecotoxicol. Env. Saf. 2019, 183, 109574. Doi.org/10.1016/j.ecoenv.2019.109574”. Even the concentration for Pb uptake in our study are slightly more but are comparable to the referenced study. There might be some reasons/mechanisms due to which reviewer thinks that there is slighter more Pb uptake by plants in our study:

In the referenced study, the Pb was applied as soil-spiking. But in current study, Pb was applied in soluble form as solution, which can be easily uptaken by plant. There is difference in soil texture of both studies. In the referenced study, the soil texture was clay loam, but in our study it is sandy loam. As the mobility of Pb is greater in sandy loam soil than that of clay loam, so there is possibility of higher Pb uptake. Experimental duration can also affect the metal uptake. In the referenced experiment, the plants were harvested after 60 days of sowing, but in ours case plants were harvested after 70 days of sowing which is 10 days more. The size of soil-containing pot might be a possible said reason. In above experiment, authors used the 3Kg soil per pot. But in current experiment, we have used the pots containing 5Kg of soil per pot. So, more soil contained more Pb. That’s why there might be more Pb uptake by plants. Phytoextraction potential varies from variety to variety. Some varieties are less but others are more sensitive against any stress. As, there are different varieties in both studies, so there is difference in Pb uptake by both varieties. In referenced study, authors have used the biochars with high pH i.e. basic. The applied biochar might have increased the soil pH and there is less uptake of metals under high pH as compared with low pH. This also could be a possible reason.

Reviewer has also commented on minimal increase in plants biomass under Pb stress in control plants(control with respect to CA). The reason behind the minimal increase in biomass is that we have applied the Pb stress for the first time after 30 days of sowing. That’s why Pb affected the said plants biomass non-significantly.

In addition, reviewer talked about the field applicability of CA. In this response, as we have applied the CA as foliar spray, so it is of greater importance that CA can be easily applicable in field condition as foliar spray. The foliar application of CA is also cost-effective and easily applicable approach.

2 line 92 (and p 10 line 294) – The application of citric acid needs to be explained more accurately and thoroughly. This is an integral component of the work and the methods used will affect the results. The application of 2.5 mM or 5 mM Citric acid only indicates the concentration of citric acid applied – what quantity of solution was applied to the soil? Was the solution irrigated into the soil, was any leachate retained in the pot and reabsorbed by the soil or was it free draining (did any citric acid leave the system)? The concentrations (2.5 mM and 5 mM) should be converted to a rate, i.e., mmol/kg soil or mmol/cm2 surface area. Applying 50 ml of 2.5 mM CA would not be the same as applying 500 mL of CA. Was Citric Acid applied as the acid form or as a sodium salt (sodium citrate).

Response

Thanks for nice suggestion. We have applied the CA as foliar spray, not is soil. We have mentioned the method used and quantity used for CA in our revised manuscript. Please see lines 312-315.

P 2 line 72 – the cited references discusses Cd uptake in Castor bean – is there an original (better?) reference for Pb uptake in shoots of Castor bean?

Response

Thanks for catching this. Relevant reference has been added to the revised references.

P 10 line 296 – “At maturity” – how are you defining maturity – is it flowering, seed set or senescence – how long were the plants grown prior to harvest.

Response

At maturity we harvested plants, it means that after 70 days of sowing we harvested. We have also mentioned the number of days in the revised manuscript. Please see line 317.

10 line 288 – how was the soils spiked – what form of lead was used. Addition of a soluble Pb salt such as lead nitrate unless allowed to equilibrate and “age” yields unnaturally high levels of soluble Pb. Additionally, as the soluble Pb hydrolyzes the soil pH will become more acidic resulting in substantially different chemical characteristics than what is presented in Table 1. I believe it very important that the soil characterization data should be presented for soil after spiking, including the actual measured concentrations of Pb in the soil after spiking – likewise the control (0 mg/kg Pb) should be spiked with a similar soluble salt so that nitrate, for example if lead nitrate was used, are at similar levels in the soil. 9, line 260 – I believe it to be widely accepted that Pb is retained on the root surface through precipitation as lead oxides, hydroxides, carbonates and phosphates, not complexed with sugars.

Response

Thanks for valuable comment. We have used the PbNO3 as a source of Pb salt. We are sorry for term “soil spiking”, as it was typing-error. We have applied Pb in soluble form as solution and mentioned in our revised manuscript. Please see lines 304-311.

Table 1 – soil characterization data presents some of the components as mmol/L – I assume this must be the concentration in the extract (saturated paste, 1:1, etc.)? – Those values should be normalized and presented on a dry soil basis (mg/kg soil, dry weight) as soluble anions

Response

Thanks for comment. Actually these are the concentrations of saturated pastes. If reviewer think still thinks so, we will change the units.

Reviewer 2 Report

The manuscript presents the beneficial properties of citric acid to counteract the effect of lead on the physiology of the castor bean plant as well as to improve the lead phytoextraction from contaminated soils. The citric acid effects were already previously reported to improve lead phytoextraction by Brassica napus, but here the authors present the effects of citric acid on soil lead uptake by a tropical plant, the castor bean.

The manuscript data are presented and discussed in a clear and organized manner.

Please find below my specific comments:

Concerning the figure 6, I understand that the data A and B are redundant with C and D. Could the authors explain why they found important to show the lead concentration (A and B) as well as the lead accumulation (C and D)? Could the authors discuss these results?

Can the authors confirm if the standard deviations or the standard errors are shown on the graphs?

The English language could be improved, especially in the discussion section.
Line 87, according to the statistics shown on the graph figure 1E, there was no significant effect at 300 mg/kg for the leaf area.

Lines 101-102, “considerably decreased” is exaggerated as according to the statistics shown in the figure 2 (A-D), there was no significant reduction of the photosynthetic pigments when Pb was applied compare to control.

Line 104, it could be added in the text that CA also has an effect on photosynthetic parameters without Pb addition (Pb0).

Line 128, the meaning of the abbreviation EL should be mentioned here in the text.

Line 132, significantly reduced compare to what condition?

Line 238, it is not the Figure 4 but the Figure 5

Line 255, it is not the Figure 5 but the Figure 6

Table 1: Please provide the meaning of the abbreviation “SAR”

Some details also need to be fixed:
Line 59, “While” could be replaced by while
Line 69, « Ricinus communis » should be in italic
Line 83, the dot after "plant" should be removed
Line 100, the dot after "gas" should be removed
Figure 3B, there is a mistake in the y axis, it should be replaced by "Photosynthetic rate"
Line 151, the dot after "plant" should be removed
Line 157, "severs" could be removed I think
Lines 160-161, the construct of the sentence should be corrected.
Line 188, "phytochelation" should be replaced by "phytochelatins" I think.
Line 220, “sometime” should be replaced by “sometimes”
Lines 225-226, a verb should be added to the sentence.
Line 233, “Mm” has to be replaced by “mM”

Author Response

Reviewer 2

The manuscript presents the beneficial properties of citric acid to counteract the effect of lead on the physiology of the castor bean plant as well as to improve the lead phytoextraction from contaminated soils. The citric acid effects were already previously reported to improve lead phytoextraction by Brassica napus, but here the authors present the effects of citric acid on soil lead uptake by a tropical plant, the castor bean.

The manuscript data are presented and discussed in a clear and organized manner.

Please find below my specific comments:

Response

Thank you for your positive comments. We would like to acknowledge the reviewer’s positive overall evaluation on our manuscript. We appreciate the reviewer for spending the time to evaluate our manuscript. All thoughtful comments have been fully addressed in our revision. We have made every effort to address reviewer’s concerns as clearly and succinctly as possible, as demonstrated in the following pages.

Concerning the figure 6, I understand that the data A and B are redundant with C and D. Could the authors explain why they found important to show the lead concentration (A and B) as well as the lead accumulation (C and D)? Could the authors discuss these results?

Response

Thank for your comment. For discussion purpose we have calculated the Pb uptake and accumulation, which shows the phytoextraction potential of castorbean. As the results showed that both uptake and accumulation increased with increasing concentration of citric acid. So, it showed that citric acid has assisted the phtoremediation potential of plants.

Can the authors confirm if the standard deviations or the standard errors are shown on the graphs?

Response

We have checked and confirmed the standard deviations on graphs.

The English language could be improved, especially in the discussion section.
Line 87, according to the statistics shown on the graph figure 1E, there was no significant effect at 300 mg/kg for the leaf area.

Response

Thanks for your comment. English language has been improved in discussion section and we have made the correction regarding figure 1E.

Lines 101-102, “considerably decreased” is exaggerated as according to the statistics shown in the figure 2 (A-D), there was no significant reduction of the photosynthetic pigments when Pb was applied compare to control.

Response

We have made and highlighted the changes in our revised manuscript.

Line 104, it could be added in the text that CA also has an effect on photosynthetic parameters without Pb addition (Pb0).

Response

We have made the changes as suggested. Please see lines 105-106.

Line 128, the meaning of the abbreviation EL should be mentioned here in the text.

Response

The change has been done as suggested in revised manuscript.

Line 132, significantly reduced compare to what condition?

Response

The significant reduction has been compared with respective control. We have mentioned it in revised manuscript. See lines 133-134.

Line 238, it is not the Figure 4 but the Figure 5

Response

Thanks for catching this. Yes its Figure 5, not the Figure 4 and we have made the correction.

Line 255, it is not the Figure 5 but the Figure 6

Response

Thanks for catching this. We have made the correction.

Table 1: Please provide the meaning of the abbreviation “SAR”

Response

We have added the abbreviation of SAR as suggested.

Some details also need to be fixed:
Line 59, “While” could be replaced by while
Response

The change has been made as per suggestion.

Line 69, « Ricinus communis » should be in italic

Response

The correction has been made.

Line 83, the dot after "plant" should be removed

Response

Thanks for catching this. The change has been made as suggested.

Line 100, the dot after "gas" should be removed

Response

Dot has been removed in revised manuscript.

Figure 3B, there is a mistake in the y axis, it should be replaced by "Photosynthetic rate"

Response

The change has been made.

Line 151, the dot after "plant" should be removed

Response

The change has been made as suggested.

Line 157, "severs" could be removed I think

Response

The word “severe” has been removed.

Lines 160-161, the construct of the sentence should be corrected.

Response

The required sentence has been revised.

Line 188, "phytochelation" should be replaced by "phytochelatins" I think.

Response

The word “phytochelation has been replaced by “phytochelatins” as per your nice suggestion.

Line 220, “sometime” should be replaced by “sometimes”

Response

Thanks for nice comment. The change has been made as suggested.

Lines 225-226, a verb should be added to the sentence.

Response

The sentence has been revised in revised manuscript.

Line 233, “Mm” has to be replaced by “mM”

Response

The unit has been corrected

Round 2

Reviewer 2 Report

The manuscript has been improved and I thank the authors for their answers.

I just have a few suggestions:

- Lines 26, 144 and 275: “caster” has to be replaced by “castor”

- Figure 3B, there is a mistake in the text of the y axis that has not been fixed in the new version submitted, it should be replaced by "Photosynthetic rate".

Author Response

I just have a few suggestions:

- Lines 26, 144 and 275: “caster” has to be replaced by “castor”

- Figure 3B, there is a mistake in the text of the y axis that has not been fixed in the new version submitted, it should be replaced by "Photosynthetic rate".

Response

Thank you very much for your time and nice suggestions. We have made the changes in revised manuscript as per your nice suggestions. Thank you once again for helping us in improving our manuscript.